# Recycling Chain for Spent Lithium-Ion Batteries

**Denis Werner \*, Urs Alexander Peuker and Thomas Mütze** 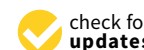

Institute of Mechanical Process Engineering and Mineral Processing, TU Bergakademie Freiberg, 09599 Freiberg, Germany; Urs.Peuker@mvtat.tu-freiberg.de (U.A.P.); Thomas.Muetze@mvtat.tu-freiberg.de (T.M.)
\* Correspondence: Denis.Werner@mvtat.tu-freiberg.de; Tel.: +49-3731-393023

**Abstract:** The recycling of spent lithium-ion batteries (LIB) is becoming increasingly important with regard to environmental, economic, geostrategic, and health aspects due to the increasing amount of LIB produced, introduced into the market, and being spent in the following years. The recycling itself becomes a challenge to face on one hand the special aspects of LIB-technology and on the other hand to reply to the idea of circular economy. In this paper, we analyze the different recycling concepts for spent LIBs and categorize them according to state-of-the-art schemes of waste treatment technology. Therefore, we structure the different processes into process stages and unit processes. Several recycling technologies are treating spent lithium-ion batteries worldwide focusing on one or several process stages or unit processes.

**Keywords:** environmental technologies; waste treatment; recycling; spent lithium-ion batteries; recycling chain; process stages; unit processes; industrial recycling technologies

## 1. Introduction

The development of information technologies, electrically powered vehicles, stationary energy storage systems, and consumer electronics will further increase the consumption of lithium-ion batteries (LIBs) over the next few years [1,2]. As a result, there will be also an increasing amount of spent batteries in the future, which will have to be treated by suitable processes [3,4]. Review articles and research activities, which focus mainly on the valuable active materials of the cathodes, define process steps involved in recovering these active materials as pretreatment. From this point of view, most of the recycling technologies, independent from the technological scale, cannot be clearly structured, represented, compared, and differentiated. Due to the increasing importance of the individual process steps prior to metallurgical treatment, this article classifies the steps for the recycling of spent LIBs along the generic process chain for waste materials providing uniform nomenclature. Moreover, it compiles an overview of the known industrial processes.

LIBs are rechargeable electrochemical energy converters in which chemical energy is converted into electrical energy by reversible redox reactions during the discharge process and vice versa during charging [5]. LIBs are one of the most important mobile energy storage devices for electrical and electronic applications. Moreover, this battery type has a great potential of success in the international market due to its beneficial properties like high energy density, no memory effect, and low self-discharge. Another fact is that the production of LIBs requires a considerable amount of metallic resources which represent a potential risk to the environment if disposed to landfills and which have to be returned to the material cycle [6].

The recycling of LIBs is of great importance not only from an economic and environmental perspective, but also from a geostrategic point of view and some health aspects [7]. Spent LIBs contain geographically unevenly distributed rare and valuable materials and generate large quantities of metal-containing waste [8]. High voltage and current from residual energy can lead to severe injuries.

Furthermore, active materials containing nickel oxide have a carcinogenic potential [9,10]. For these reasons, recycling of spent LIBs enhances environmental protection and the idea of a circular economy by separating the valuable metallic constituents into different products. These products are secondary raw materials for the production of metal or metal composite products [3].

Besides the intrinsic value of the battery materials and components, national or supranational legislation like the Battery Act or the Directive 2006/66 EC provide the framework for the recycling of LIB. These directives prescribe responsibilities and procedures for each contributor included in the life cycle of a battery. Moreover, the directives classify different battery types in respect to their implemented application and address waste management. Especially, collection and recycling targets are formalized to promote material instead of energy recovery or disposal. Therefore, recycling is defined as the reprocessing of waste materials in a production process either for their original purpose, i.e., material or raw material recovery, or for other purposes, i.e., other recovery like backfilling, but excluding energetic recovery [11]. Recycling efficiency is the result of the ratio of output material to feed material and targets 50 mass percentage. However, the different purposes of the output materials are not further distinguished. Moreover, recycling efficiency can be already achieved by a beneficial ratio of the cell mass in a battery system, when the battery system is dismantled to cell level.

Since their market launch in 1990, the energy and power densities of LIBs increased continuously and new areas of application have been opened up [12]. This led to a high and still increasing variety of battery types with different material compositions [13]. In this initial stage, robust pyrometallurgical recycling technologies are mainly used on industrial scale for waste treatment. These technologies reach the recycling efficiency by backfilling the slag. In addition, operating above the smelting temperatures of the contained metals pyrometallurgical technologies are very energy-intensive at the same time [14], since they are mainly focusing on high value materials like cobalt, nickel and copper. Therefore, and with regard to the increasing number of battery types and the global energy consumption, international research also focuses on the development of alternative or supplementary mechanical processes in combination with hydrometallurgical refining [15,16].

## 2. Material Recycling of Lithium-Ion Batteries

The smallest functional unit of a LIB is called a battery cell. It generally consists of two electrodes, a separator, electrolyte, and cell housing. Each electrode is composed of a metallic conductor foil and a coating, the so-called active material. By definition, the designation of the negative and the positive electrode is assigned during the discharge process. Therefore, the anode is mostly a copper foil with graphite coating and typically, the cathode is an aluminum foil coated with an intercalated lithium compound. The separator is mostly a porous polyolefin, the electrolyte a mixture of organic solvents and a lithium salt. The cell housing is a sealed container made of aluminum, steel, special plastics, or highly refined aluminum composite foils [17]. Battery cells are cylindrical, prismatic, or bag-shaped. The cells can be connected in series or parallel to form either a single block or a module as subunit of a bigger battery system [5,9,18].

If the recycling chain for waste materials is applied to LIBs, the treatment has to be subdivided into process stages on the one hand which are characteristic in terms of process technology and into unit processes on the other hand which limit the amount of waste material and the material conversion processes. Therefore, the recycling chain consists of four process stages with two unit processes each [19,20]:

1. Preparation: waste logistics and presorting,
2. Pretreatment: dismantling and depollution,
3. Processing: liberation and separation,
4. Metallurgy: extraction and recovery.

Typical recycling processes do not mention the process stages' preparation and pretreatment [21,22]. However, both of them have a significant influence on the efficiency of the downstream steps, i.e.,

processing and metallurgy. Furthermore, the differentiation between recycling concepts becomes more precise if their combinations with preparation and pretreatment is indicated as well. As a result, both process stages are included into the holistic characterization of the recycling processes of spent LIB here.

### 2.1. Process Stages of the Recycling Chain for Lithium-Ion Batteries

Figure 1 shows the generic recycling chain for lithium-ion batteries with its four process stages and the associated unit processes. Different mixtures of batteries are collected and sorted in the preparation stage either according to battery types and/or to active materials. During the subsequent pretreatment and processing, the first secondary raw materials are produced to be fed into established recycling processes. The treatment of enriched fractions e.g., concentrates of active materials, occurs within the framework of metallurgy. The recovery of organic solvents as well as electronic and auxiliary components or casing and support materials, i.e., plastics and metals, are not considered in detail, here (cf. Figure 1). For most of these materials conventional recycling chains already exist.

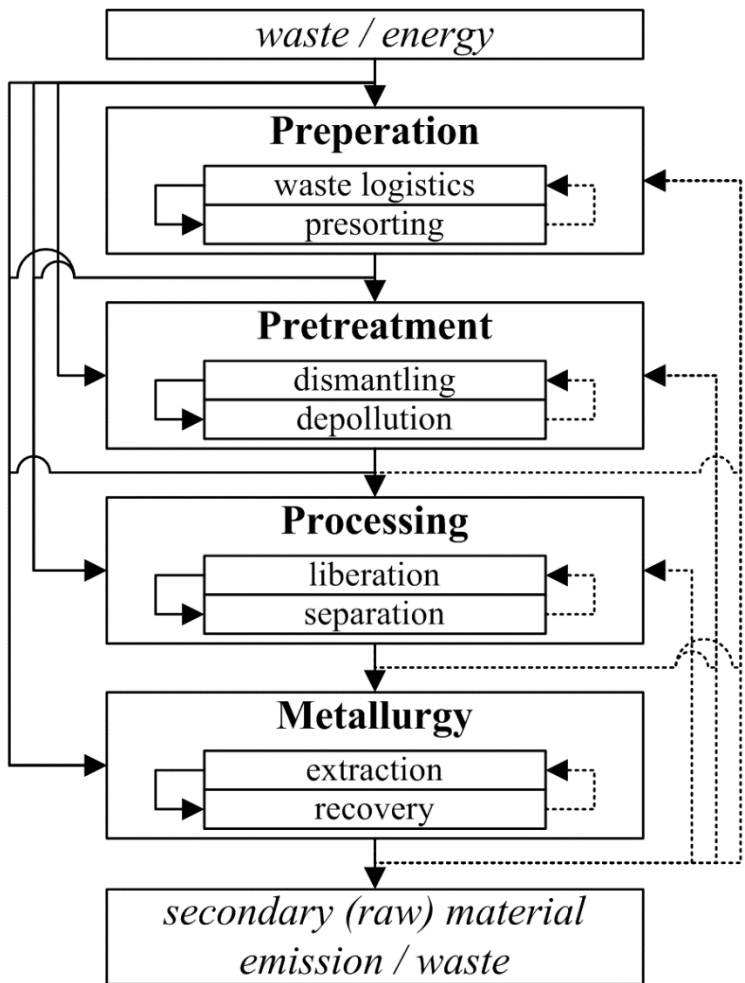

**Figure 1.** Recycling chain with process stages and unit processes for spent lithium-ion batteries.

Since it focusses on the central challenges of battery recycling, this generalized recycling chain can be used to evaluate all of the present recycling concepts, as well as in laboratory scale to whole industrial facilities. Some of the unit processes within the process stages are used iteratively, others are omitted altogether (cf. Figure 1) [20].

## 2.2. Unit Processes

The individual unit processes are composed of unit subprocesses and unit operations that are linked in different complexes. Unit subprocesses describe the character of the implicated state of aggregation, whereas unit operations treat a material within a single device [23,24]. Martens and Goldmann [19] present the general objectives of the respective unit processes in detail. These are explained below for spent LIB.

### 2.2.1. Preparation

The unit processes of preparation as the first process stage in LIB recycling are waste logistics and presorting as shown in Figure 2.

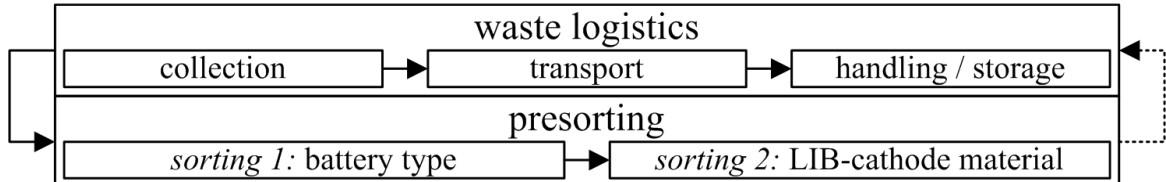

**Figure 2.** Process stage 1 for recycling of lithium-ion batteries: preparation.

### Waste Logistics

In the past, LIBs were mainly produced for portable and mobile applications. Currently, different collection systems collect the batteries depending on the legislation of the respective country [25,26]. For example, according to the European Directive (2006/66/EC) on waste batteries and accumulators, distributors of industrial batteries from electric vehicles have to take back the batteries themselves. Therefore, collection systems for those battery types are either already available or under development. In this field, car manufacturers do often closely cooperate with local recycling companies due to the low sales figures and correspondingly few returns now [27–31]. Afterwards, spent LIBs are either transported directly to the subsequent treatment facility or handled and stored before they are handed over [32,33].

Nevertheless, there is a great potential for the collection of spent LIB. One example is the LIB collection rate of Germany for portable systems e.g., smartphones or laptops which amount to only 45%. In a global comparison, even this figure in many countries is not achieved and more often no collection systems exist [34,35]. However, a large proportion of portable batteries are subject to the hoarding effect [36] or stockpiling [9,37]. Besides, they end up in incinerators, landfills, or other wastes due to a lack of consumer or sorting personal awareness, respectively [29,37–43]. In addition, electrical and electronic scrap containing LIBs is illegally exported from the developed to African and Asian countries, which also reduces the collection rates [44,45].

Due to the considerable high hazard potential (cf. Depollution), both the collection and transport of LIBs require special measures against short circuits and leakage of the electrolyte [46]. The classification of spent lithium-ion batteries as a hazardous good [13] demands special transport containers, warning signs, and packaging [46–52]. Therefore, the waste logistics cause a significant share of the total cost in battery recycling due to the high safety requirements and the resulting low specific transport weight [53].

### Presorting

Spent batteries of any origin are usually not collected separately. They are accumulated as battery mixtures of different battery types or of lithium-ion batteries of different composition [29,36,54]. Due to the large number of battery types, it is not possible yet to recover all the constituents from the mixture using a single recycling process [26,55,56]. Therefore, presorting by battery type is necessary for recycling technologies specializing in lithium-ion batteries to define material flows for further treatment.

The sorting technologies used for presorting of batteries are assigned to picking. Characteristic features of each individual battery are analyzed and evaluated using trained personnel or sensors. Then, the batteries are separated manually or automatically [29,47,55]. Recycling technologies based on picking are designed for complex battery materials, so high recycling rates can be achieved [34,56]. In addition, it is possible to sort by different cathode active materials (e.g., Figure 2) within a single LIB type. However, picking requires an initial dismantling (cf. Dismantling) of the battery cells into their functional unit and an elemental analysis of the active material previously. As this type of preparation is associated with a significant processing time and hazard potential (cf. Depollution), it is currently not used industrially.

Currently, most of the technologies for recycling of spent LIBs operate without presorting, especially technologies based on pyrometallurgical unit operations [16] (cf. Pyrometallurgy). The processes were derived from the recycling of nickel-based battery systems [57,58], the production of primary and secondary metallic raw materials [16], or the recycling of completely different wastes [35]. The changing composition of these mixed material systems causes low recycling rates and poor quality of the secondary raw materials. Therefore, only metals like nickel, cobalt, and copper are recovered and by-metals like aluminum or manganese are discharged into the furnace slag [34]. The latter ones can only be reintroduced into the material cycle at great expense.

### 2.2.2. Pretreatment

After the preparation stage, the batteries need to be dismantled to a defined level. Furthermore, different hazard potentials are deactivated thermally, electrically, or cryogenically (cf. Figure 3). Depending on the size or original purpose of the battery system, both linear and iterative pretreatment is performed within this process stage and its unit processes.

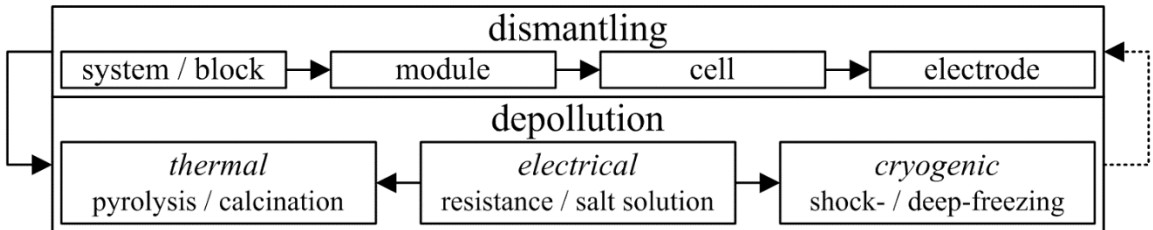

**Figure 3.** Process stage 2 for recycling of lithium-ion batteries: pretreatment.

Dismantling

The increasing market penetration of electric cars and the resulting increase of spent LIBs in respect to amount, mass, and size makes dismantling reasonable and in some cases urgently necessary. Dismantling is a time-consuming and thus cost-intensive process step due to the complexity of spent batteries [9,56]. Often, technological equipment such as furnaces or crushing devices are limited regarding to the maximum size or mass of the feed in order to achieve process ability, sufficient process stability, and efficiency [59]. In addition, it is indispensable for some methods of depollution to access the battery modules or cells (see Depollution). Besides the battery cell, a battery system consists of a high proportion of periphery components like the battery management system (BMS) or cooling parts. Therefore, dismantling of LIBs from system to module, cell, or even electrode level generates products made of metals, plastics, and electronic components that can be fed to established recycling routes increasing the overall recycling efficiency [14,56,60,61]. Furthermore, functional components or reusable assemblies can also be obtained for Second Life applications [9,62]. From an economic point of view, dismantling is an optimization problem between the dismantling level, i.e., the revenue from recovered individual parts or energy savings in mechanical and/or metallurgical processing, and the costs for the equipment and operating expenditure [9].

Several dismantling concepts are known, such as manual, semi-automatic (hybrid), and fully automatic dismantling [9,56,63]. From the economical and safety perspective, it is difficult to implement only manual dismantling [54]. Hybrid concepts try to overcome this by combining manual activities with robots [64]. The implementation of automated dismantling by industrial robots is subject of controversial discussions about the manifold battery designs and the rapid technical development of batteries [9,46,54,56,64–66]. Currently, battery systems are manually dismantled to different dismantling levels within the framework of research projects and some industrial applications [39,54,66,67]. Occupational safety requires that the DC voltage is reduced at least below 60 V [54,68].

Depollution

The depollution within the recycling of LIBs prevents carry-over of critical or hazardous components into subsequent process steps and avoids the release of harmful emissions into the environment [19]. The hazard potentials of spent LIBs can be summarized as electrical, chemical, and thermal ones which interact with one another [56,69]. Depending on the individual recycling process, its intermediate and final products, the depollution utilizes different methods like discharge, cryogenic treatment, and/or thermal treatment to remove hazardous substances or conditions [70]. As a result, depollution is also called as deactivation [10], passivation, or stabilization [9].

Discharging is a method that lowers the electrochemical energy content of the battery [9]. This method is primarily used for recycling processes using dismantling (cf. Dismantling) or mechanical liberation (cf. Liberation) [31,71]. Various methods for discharging were subject of the research project LithoRec II [10,18,63], whereas discharging using an external circuit with resistor is the most common and a practical method for large battery cells with high capacities [16,71]. In addition, cells are discharged in salt brines [9,72–74], in powders from metallic conductor foils or graphite [75], or in stainless-steel containers with stainless-steel chips [70]. Especially, when using brines, undesired side reactions are also mentioned leading to corrosion of electrical contacts or housing components as well as to the release of hydrogen [10,31,75] or other gases [9]. The discharge in salt brines (mainly NaCl) in particular is currently a common method for discharging for low capacity batteries [70,74]. If high voltage batteries are discharged, this method can lead to leaks in the battery housing and has its limits regarding to discharging process and efficiency as well as contamination of electrode materials [75]. According to Zhao [31], only high discharge currents and professional technical equipment can achieve the safe and complete discharge of LIBs. However, this results in an enormous expenditure of time and money, especially with regard to high-performance batteries from electric vehicles [9]. This does not apply to batteries with internal damage anyway.

Cryogenic treatment is one method that avoids exothermic reactions, especially during liberation (cf. Liberation). If LIBs are exposed to temperatures around −200 °C, the ion mobility decreases significantly [10,69]. Appropriate safety measures must be taken (e.g., heat-resistant conveyor units and exhaust gas purification systems), since the chemical reactions after liberation occur later compared to the treatment at normal temperatures.

Thermal methods such as pyrolysis or calcination easily remove flammable electrolyte components and decompose the electrolyte components by breaking down the organic compounds thermochemically. The fact that these thermal processes partially decompose components such as the separator and the binder of the electrode coatings and that they delaminate the coating from the metallic conductor foils is in turn used in a number of processes [54,76]. These processes are carried out in vacuum induction furnaces [76], rotary kilns [34], or blast furnaces [77]. Appropriately designed dedusting and flue gas cleaning systems then separate the resulting decomposition products [6,10,54,76]. At laboratory scale, pyrolysis in tube furnaces with a vacuum environment is also applied as pretreatment of mixed active materials for further lithium recovery [78].

### 2.2.3. Processing

The third process stage in the recycling chain of LIBs is the processing with the unit processes liberation and separation (cf. Figure 4). The aim of this process stage is to break up the bonds between the individual components, i.e., materials, in order to separate them into defined concentrates. Liberation also includes size control to influence adaptability and efficiency of physical separation technologies. Especially, if the active materials are separated from the metallic conductor foils, impurities must be minimized in the corresponding fraction in order to unburden or even enable the subsequent refining (cf. 2.2.4) [3].

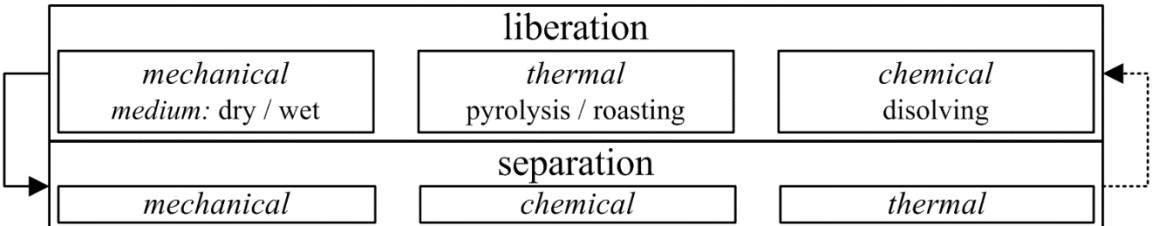

**Figure 4.** Process stage 3 for recycling of lithium-ion batteries: processing.

A separation efficiency depends generally on the mechanical degree of liberation of the individual components that need to be separated [79]. Secondly, it depends on the efficiency of the separation technology used. A combination of crushing, size classification, and sorting is intended to either produce secondary raw materials in the sense of material recycling, to prepare the material recycling by metallurgical processes, or to achieve waste treatment and disposal [33]. In addition, thermal, chemical, and mechanical separation processes can be used for special purposes e.g., electrolyte separation by drying, de-coating of the metallic conductor foils, or shape modification of crushed and enriched materials [18,80], respectively.

Liberation

The mechanical liberation of LIBs is mainly achieved by shear, cutting, and tearing stresses since their material behavior is ductile [81]. Therefore, mechanisms based on slow or fast compression are more likely to trap materials or create new compounds [82]. The liberation of LIB modules is usually carried out in two stages [34,82], rarely in one stage [10,16,54,83]. The first step is precrushing of the feed and the second liberation of the components itself. During this mechanical treatment it is essential to protect the tools and process chambers of the equipment against corrosion caused by electrolytes [31]. Furthermore, there are approaches for electrohydraulic defragmentation of LIBs, but so far, this could not be converted to a large technical scale successfully [84–87]. Liberation by using diamond saws, water, or laser beams is described in the literature as well, but it is very time consuming and lacks efficiency [31]. In addition, there is always a fire hazard when diamond saws and laser beams are used.

Depending on the type of pretreatment and depollution (cf. 2.2.2), it is necessary to adopt adequate safety measures to provide a safe work environment during the liberation. For example, the medium in the process chamber can cause strong exothermic decomposition reactions up to explosions [3]. Ideally, each single battery cell is completely discharged for liberation [20,31,80]. The hazard potential deriving from volatile electrolyte components can be reduced by thermal or cryogenic treatment in order to use ambient air during crushing. Charged low capacity batteries have to be crushed either under a protective gas [3,9], such as carbon dioxide [88], nitrogen [63], argon [88,89], or helium [69], or in liquid media, such as water or salt solutions [88]. Using water, further safety precautions are required due to undesirable side reactions of the electrolyte with the process medium (e.g., formation of hydrofluoric acid). Regardless of the process medium, the volatile electrolytes and dust must be separated from the medium afterwards by adequate purification systems.

Thermal and chemical liberation methods are applied to delaminate the coating material from the current collector foils, especially for cathodes. Thermal methods like pyrolysis or roasting decompose the binder, whereas chemical methods either dissolve the aluminum or detach the cathode active materials from the aluminum foil in a special solution [9].

Separation

Mechanical sorting according to electromagnetic, electrostatic, density, and granulometric properties mainly separates liberated components and materials. At times, hydrometallurgical treatment follows separation, hydrometallurgical processes require a high purity of the intermediate products for sufficient process stability and selectivity. Commonly used processes are magnetic and eddy current separation, screening, gravity sorting in flow fields [31,69] or with pneumatic shaking tables [88] as well as flotation, to either enrich valuable materials or to deplete impurities in fractions [3,88,90]. Furthermore, it is necessary to install appropriate measures for the dedusting and separation of the electrolyte from the process media, latter thermally or chemically [18,31].

The combination of separation steps is extremely material and process specific. Nevertheless, in most recycling processes adopting processing technologies the materials are enriched in products such as casing materials, plastics (mainly the separator foil), a mixture or separated electrode foils, and a mixture or the individual active materials [9,18,80,91]. If the electrode foils still contain active materials after the mechanical liberation [9], coating and metallic foil are liberated and separated by further mechanical [14,80], thermal [92] or chemical treatment [3,77,93,94] (cf. Liberation).

Active materials can be sorted by flotation [95] or multistage magnetic separation after appropriate preparation steps [96]. This mechanical processing of active materials is the preferred option from the energetic and geostrategic point of view (cf. 1). Challenges arise from the small particle sizes at sufficient liberation and the small differences in the material properties there [79]. Therefore, the active material fractions are mostly not sorted mechanically but fed to either established or newly developed pyro- and/or hydrometallurgical processes (cf. 2.2.4).

2.2.4. Metallurgy

The final stage of the LIB recycling feeds pretreated, i.e., dismantled, batteries or material fractions from processing into extractive metallurgical processes for the production of sufficiently pure intermediate materials. The unit processes are extraction and recovery. Both unit processes are always carried out in a coupled manner. Consequently, the refining process is classified into hydro- and pyrometallurgical processes (cf. Figure 5). For materials used in the housing of the batteries, in peripheral components, and the conductor foils established metallurgical processes exist, e.g., for aluminum, copper, and alloyed steel. These components are treated in pure secondary smelters using specifically designed and optimized processes [19]. The metals cobalt, nickel, copper, manganese, and iron bear an outstanding position in the recycling of LIBs due to their high intrinsic value and the comparatively low cost of recovery [56]. Here, pyrometallurgy can be combined with hydrometallurgical processes but pure hydrometallurgical processing is becoming more and more important [39,58,97,98]. When discussing different metallurgical technologies for the processing of spent LIBs, the pyro- and hydrometallurgical unit processes are compared as stand-alone.

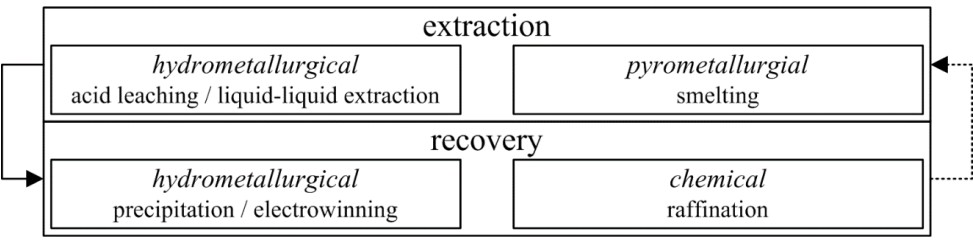

**Figure 5.** Process stage 4 for recycling of lithium-ion batteries: metallurgy.

Pyrometallurgy

In industrial pyrometallurgical processes, the solid feed is melted down, typically in an electric arc furnace [99] or shaft furnace [7], and transferred into an alloy, a slag, and/or a matte and the off gas with dust particles [9]. Organic components of the feed are pyrolyzed or entirely burned depending on the furnace technology. Graphite can be generally used as a reducing reagent for carbo-reductive melting processes. If the graphite content of the feed material is too high, problems occur during pyrometallurgical processing influencing reaction kinetics due to low reactivity of graphite, properties of the melt like melting point and viscosity, slag formation, and consequent metal recovery as well as the overall process efficiency [7,99]. Depending on the composition of the feed and the process specifications, cobalt, nickel, copper, and iron can be collected as metal alloy or matte. The metal alloy is sold as a secondary raw material, such as alloyed steel [100], whereas the matte is further processed hydrometallurgically. Unfortunately, due to its principle, pyrometallurgical processes transfer lithium, manganese, and aluminum into the slag which is currently used as filler material, e.g., in road construction [34,37,92] or in concrete [101], or is deposited in landfills [40]. In principle, the recovery of these three substances from slag is technically feasible via hydrometallurgical processes [9,40] but not economically [6,35,88,102,103]. Furthermore, pollutants such as carbon dioxide, dioxins, and furans occur [3,9,93,104].

Pyrometallurgical technologies exhibit comparatively low flexibility due to the required high economic investments and the complex processes involved in the extraction of metals [93]. Other disadvantages of such technologies are low capacities, high energy consumption, as well as limited recycling efficiency [9,37,93]. An advantage of this technology is its robustness which requires only minor pretreatment and conditioning of the feed since many hazard potentials are eliminated automatically during smelting [9].

Hydrometallurgy

One option in pyrometallurgical processing is to produce intermediate instead of final products which are refined by hydrometallurgical processes [105]. Currently, the favorable option is to directly process active material concentrates with a combination of several hydrometallurgical unit operations [16,70]. Hydrometallurgical processing is mainly applied for the metals coated on the cathode. The processing can be subdivided into dissolving and concentrating of the feed as well as cleaning and recovering the metal salts [106]. Various organic and inorganic acids were examined as solvents often mixed with deoxidizing agents to increase the recovery rate [9,93,106]. Furthermore, bio-organisms are able to dissolve and convert the metals by caustic methods as well [4,9,77].

Once dissolved, the metals are extracted from the solvent by liquid–liquid extraction, ion exchange, or chemical precipitation [69,77,97]. If the resulting metal salts meet the quality requirements of the corresponding raw materials, subsequent recovery can be forgone. Otherwise, further precipitation, crystallization, or electrochemical processes such as extraction electrolysis or electro winning are used. Thereby, corresponding metal compounds or impurities are separated selectively or deposited on electrodes, respectively [4,77,106].

Hydrometallurgical processes require material preparation and size control provided by manual dismantling and/or liberation and separation. The direct manual dismantling of batteries to electrode level provides highly pure feed materials on one hand [61] and decreases the required piece sizes for further hydrometallurgical processing on the other. For industrial implementation, however, manual removal of the valuable cathode materials does not appear to be expedient from both an economic and occupational safety point of view [70] (cf. Dismantling and Depollution). If the electrode coatings are separated mechanically (cf. 2.2.3), the coating fraction contains some copper and aluminum particles of the metallic conductor foils as contaminants [9,80]. Their particle size distribution and material composition are thus significantly influenced by the type of processing, which in turn determines the metallurgical effort and yield of recyclable materials [9,106]. Further impurities are residues of organic solvents, which influence the pH value of the solution and the performance of other solvent based

processes. Therefore, these impurities have to be considered in the design stage of hydrometallurgical processes as well.

As a more general rule, hydrometallurgical processes require strong organic acids and expensive additives producing considerable amounts of waste liquids [107] and harmful or toxic emissions [22]. In contrast, metals from the cathode coating can be recovered in an energy-efficient and selective way [4,69].

## 3. Industrial Recycling Technologies

Numerous recycling companies worldwide treat spent LIBs of different types and forms (cf. Table 1). The recycling technologies differ according to the used process stages and unit processes as well as the generated final products. This is due to the historical development of the individual companies, the environmental conditions and regulations, as well as the relevant market situation. In principle, it is possible to characterize each technology on the basis of the process stages and unit processes introduced in Section 2. Literature often shows only the unit processes for the last process stage (metallurgy) in tabular form (cf. Table 1—data from tables). In some cases, there are still indications of pretreatment and mechanical processing from which the overall recycling technology can be guessed, but which cannot be reliably traced. Detailed descriptions and explanations are available for certain process technologies (cf. Table 1—process described). However, the low information density prevents a clear and complete characterization according to the process stages and unit processes [6].

**Table 1.** Overview of industrial recycling technologies for spent lithium-ion batteries (mech = mechanical processing; hydro = hydrometallurgy; pyro = pyrometallurgy; n. d. = no data).

| Company | Data from Tables | Process Described |
|---|---|---|
| ACCUREC GmbH | mech [108], pyro [3,26,69], pyro and hydro [10,59], pyrolysis and hydro [109,110], disassembly, pyrolysis, mech [7], n. d. [111,112] | [3,7,10,13,98,113] |
| AEA Technology Batteries | hydro [59,69] | [98,114] |
| AERC Recycling Solutions | pyro [59] n. d. [69] | - |
| AFE Group (Valdi)/ ERAMET | pyro [59,69] | - |
| AkkuSer | mech [59], mech and hydro [110], n. d. [111] | [98,113] |
| American Manganese | n. d. | - |
| Anhua Taisen Recycling Technology Co. Ltd. | mech and hydro | - |
| Battery Resourcers LLC | n. d. | [13] |
| Battery Safety Solutions | collection, discharge and disassembly | - |
| Batrec Industrie AG | mech [108], mech and pyro [59], mech and hydro [26], pyro [3,69], hydro [10], pyrolysis and pyro [109,110], mech, pyrolysis, mech, hydro [7], n. d. [111,112,115] | [3,7,10,13,98,113,115] |
| BDT | n. d. [111] | - |
| Brunp Recycling Technology Co. | hydro [3,59,116] | [13] |
| Cawleys | n. d. | - |
| Chemetall | n. d. [1,2] | - |
| DOWA Eco-Systems Co. Ltd. | pyro [59,69], n. d. [111,115] | [117] |
| DK Recycling und Roheisen GmbH | pyro [69] | - |
| Düsenfeld GmbH | mech and hydro [118] | (for LithoRec [13]) |
| Earthtech | disassembly | - |
| Erlos/Nickelhütte Aue | disassembly [112], pyro and hydro | - |
| Euro Dieuze Industrie/ SARP | hydro [59,69,109,110], n. d. [111] | - |
| Farasis Energy | - | [13] |
| Fuoshan Bangpu Ni/Co High-Tech Co. | n. d. [111] | - |
| GHTECH | - | [13] |
| G&P Batteries (Ecobat Technologies Ltd.) | pyro and hydro [109,110], n. d. [59,111,115] | - |
| GRS Batterien | pyro [109,110], n. d. [111] | - |
| Guangdong Guanghua Sci-Tech Co., Ltd. | disassembly | - |
| Highpower International Inc. | disassembly, pyro and hydro | [13] |
| Huayou Cobalt New Material Co Ltd. | pyro | - |
| Inmetco | pyro [3,7,69], n. d. [115] | [7,10,13,113,119] |
| Japan Recycling Center | pyro [69] | - |
| JX Nippon Mining and Metals Co. | pyro [59], pyro and hydro [69], n. d. [111] | - |
| KYOEI Steel | pyro | - |
| Li-Cycle US | mech and hydro | - |
| Lithion Recycling | hydro | - |
| Metal-Tech Ltd. | n. d. [111,115] | - |
| Neometals | hydro | - |

**Table 1.** *Cont.*

| Company | Data from Tables | Process Described |
|---|---|---|
| Nippon Recycle Center Corp. | pyro [59] | - |
| OnTo Technology Oregon US | hydro [69], mech [116], presorting, disassembly and hydro [7] | [7,13] |
| Pilagest | mech and hydro [109,110], n. d. [69] | - |
| PROMESA GmbH & Co. KG | mech | - |
| Recupyl S.A.S | mech [108], mech and hydro [7,110], hydro [3,10,26,59,69,109], n. d. [111,115] | [7,10,13,98,113,119] |
| REDUX GmbH | pyro [69], pyrolysis and mech | [54] |
| REVATECH | n. d. [115], n. d. [111] | - |
| SAFT. AB | pyro [69] | - |
| Salesco Systems | pyro [69] | - |
| Shenzhan BAK Battery Co. | disassembly [31] | - |
| Shenzhen Green Eco Manufacturer Hi-Tech. Co., Ltd. | mech and hydro [116], hydro [3,59], n. d. [111,115] | [3,13] |
| Shenzhen Tele Battery Recycling Co. | hydro [31] | - |
| SK Innovation Co | n. d. | - |
| S.N.A.M. | mech, pyrolysis and pyro [110], pyro [3,69,109], pyro and hydro [59], n. d. [111,115] | [98,113] |
| Sony Corp. & Sumitomo Metals and Mining Co. | pyro [3,59,69], n. d. [111,115] | [3,13,98,113] |
| Soundon New Energy Tech. Co. Ltd. | - | - |
| SungEel Hitech Ltd. | mech and hydro [116] | [13] |
| Technologies Inc | n. d. [59] | - |
| TES-AMM China | n. d. [111] | [13] |
| Toxco/ Retriev Tech. | hydro [26,59], mech [69,108], disassembly, cryogenic pretreatment, mech and hydro [3,7,10], n. d. [111,112,115] | [3,7,10,98,113,115,119] |
| Umicore | pyro [10,59], pyro and hydro [3,7,26,108–110], n. d. [69,111,112,115] | [3,7,10,13,98,113,119] |
| Xstrata/ Glencore | pyro [7,26], pyro and hydro [3,59,69,108], n. d. [111,115] | [7,10,13,98,113] |
| 4R Energy Corp. | n. d. [116] | - |

The production of LIBs has so far taken place almost exclusively in China, South Korea, and Japan [39,58,77,120]. Hence, battery waste is mainly recycled in Asian and only a few European and North American plants [26,29,82,91,111,121]. American and European recycling companies show a wide variety of technologies but lack the volumes of spent batteries for profitable operation [70,121]. Avoiding high investments for dedicated process equipment, spent LIBs are also fed as secondary feed in existing metallurgical plants [7] (cf. Presorting).

Lv et al. [6] and De-Leon [116] published information on the capacities of specialized technologies for certain feed materials or material mixtures. Currently, the industrial recycling approaches focus primarily on the recovery of the valuable metals cobalt and nickel from portable and industrial batteries [29,40]. Therefore, the entire LIB is broken down either thermally or mechanically in order to be recovered by pyro- and/or hydrometallurgical processes. Aiming to increase the total recycling efficiency of current industrial recycling processes, also aluminum and organic battery components, such as the electrolyte and plastics, should be considered for recovery [20,80,99]. Hence, unit operations like thermal pretreatment or separation have to be adjusted in order to avoid decomposition of the organic battery components.

In general, three industrial process routes can be identified for material recovery of spent LIBs depending on the temperature depolluting the batteries, effort for preparation and processing and overall recycling efficiency (cf. Figure 6).

- high temperature route with optional presorting and calcination as deactivation, no processing but direct pyrometallurgical treatment (cf. Figure 6A)-p), and optional hydrometallurgical refining (cf. Figure 6A)-h)
- moderate temperature route with pyrolysis as thermal pretreatment, multistage mechanical processing (cf. Figure 6B)-n; n ≥ 1), and pyro- and (cf. Figure 6B)-p-h)/or hydrometallurgical (cf. Figure 6B)-h) refining
- low temperature route (often called direct recycling process [13]) with electrical and no/or cryogenic depollution, multistage processing, and hydrometallurgical refining (cf. Figure 6C)-n; n ≥ 1)

In general, the material, characterized by the arrows with continuous lines, flows top down through the different process stages and unit processes transforming spent batteries into secondary (raw) materials, emissions, and waste. Arrows with dotted lines show on the one hand an optional hydrometallurgical treatment after pyrometallurgical unit operations (cf. Figure 6A,B) or an optional

iteration of the respective unit operations. Due to generalization and low information density of the industrial technologies, the material transformation within the process stages and unit processes cannot be further distinguished in concentrates for further treatment and secondary materials.

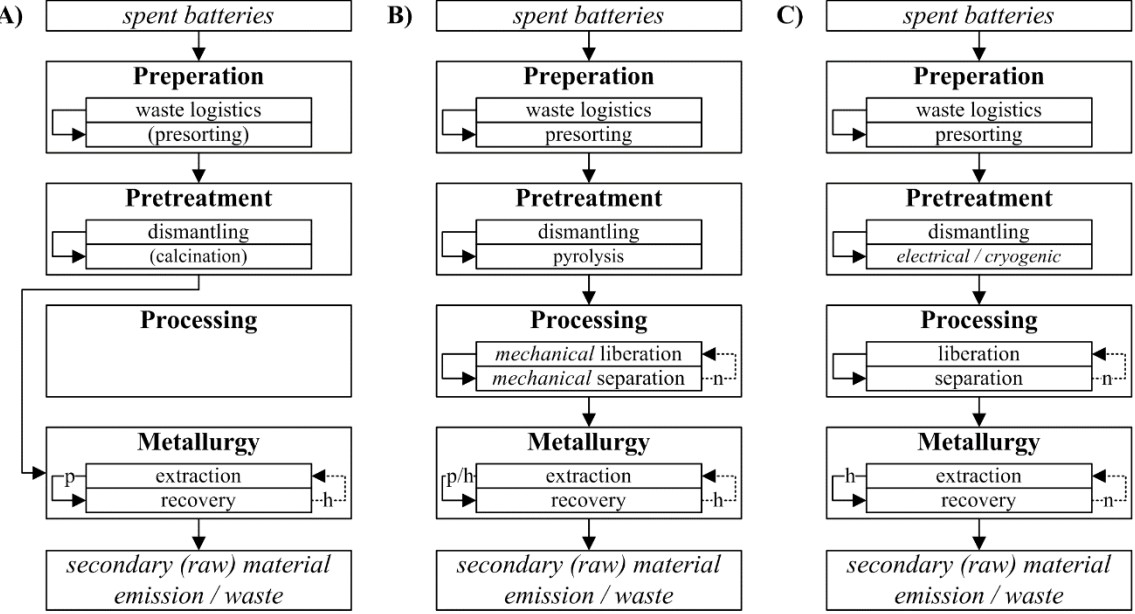

**Figure 6.** Schematic flow chart of the three most common industrial process routes: high temperature route (**A**), moderate temperature route, (**B**) and low temperature route (**C**); p—pyrometallurgical, h—hydrometallurgical, n—number of iterations.

It can be stated that the possible overall recycling efficiency decreases with higher process temperatures due to the decomposition of organic components and later due to the downgrading of metals like aluminum and manganese into slag. A detailed discussion of the different process routes is presented by Harper et al. [9], Lv et al. [6], Chen et al. [13], and Pinegar and Smith [7]. Nevertheless, an overall quantitate process analysis for the three main and other industrial recycling technologies in respect to recycling efficiency, energy, and auxiliary material consumption are scarce in the literature [101]. For example, the energy demand to recover the different metals depends on the specific recycling technology employed, as well as the forms of the final secondary (raw) materials [101]. Only one life cycle assessment can be found in literature for technologies containing two process routes. On the one hand, a combination of the unit processes dismantling, depollution, pyrometallurgical extraction, and recovery represents the high temperature route [60], and on the other hand a combination of the unit processes dismantling, depollution, liberation, separation, multistep hydrometallurgical extraction, and recovery [122] characterize the low temperature route.

## 4. Conclusions

The global search of recycling technologies for spent LIBs has led to various nomenclature and confusing classifications of relevant processes. The common goal is a high recycling efficiency and the suitability in terms of recovery of material and energy. In contrast, the generic process chain for waste from Martens and Goldmann [19] offers the possibility of dividing the individual processes into four process stages and associated unit processes. The different recycling technologies can thus be clearly classified, differentiated, and process specifications addressed.

It is conspicuous that, currently, no company is carrying out the entire process chain for spent LIBs. They rather specialize in certain process stages, combinations of process stages, or only unit processes. Advantages and disadvantages of the unit operations discussed in this review address the existing technologies knowing that an optimized entire process chain does not exist yet. Most probably,

the optimum from an energetic, material, ecological, and economic point of view will be found from a combination of different unit processes and operations. It is noticeable that some of the unit processes are partly iterated. For example, the process stage of processing can have multiple steps of liberation and subsequent separation. Then, other stages or groups are completely skipped, especially in technologies with pyrometallurgical processes, which partly dispense with depollution or processing. Although further and deeper information about the industrial technologies are missing, a rough assessment and understanding of these technologies can be gained in respect to the currently used process stages and unit operations. Nevertheless, a reliable comparison and evaluation of the several recycling technologies cannot be done.

So far, research has focused on metallurgical processing with pyro- and increasingly hydrometallurgical processes. On a laboratory scale, manual dismantling of the battery systems and subsequent dismantling of the battery cells to electrode level delivers usually the relevant feed materials (cathode coating). Extrapolating this to industrial plants, mechanical liberation with subsequent sorting will become necessary. Different approaches are available for this mechanical treatment. However, they are not distinguishable based on their unit processes alone. In such a case, the unit processes have to be further subdivided into unit subprocesses and unit operations like liberation dependent on the stresses applied (kind, intensity, speed of stressing tools) process medium, specification of the machines, and apparatus used, etc. [88]. The same applies to the other process stages, but primarily hydrometallurgy, where parameters such as solid–liquid ratio, solvents used, and process conditions (temperature, residence time) influence the material conversion processes. Necessary information is also lacking due to the economic competition situation and the current spirit of optimism that will not change in the near future.

The article shows that production of secondary raw materials or materials from spent batteries is in principle technically possible. Moreover, waste treatment of spent LIBs is currently carried out already worldwide [13]. Though the collection of spent batteries and the revenue-generating sale of the secondary raw materials produced remain decisive for the recycling [123]. Additionally, the political framework for an economically and ecologically reasonable collection system need to be created for long-term success [9]. Finally, recycling of spent LIB can contribute to the global supply of metallic resources for LIB production in the long term, but for the ongoing increasing demand and battery wastes, primary resources will stay inevitable [124] and present global recycling capacity has to be expanded [70], respectively.

**Author Contributions:** T.M. created the initial idea for the article. D.W. produced the original concept of the Review, and wrote the article, integrating contributions from T.M. and U.A.P., editing and shaping the Review. T.M. and U.A.P. contributed to the nomenclatures and critically revised the article, the former multiple times. All authors have read and agreed to the published version of the manuscript.

**Funding:** This research received no external funding.

**Acknowledgments:** The author would like to acknowledge the technical support and contributions in kind of materials used for prior experiments by BMW Group Munich, especially Recycling und Demontagezentrum of BMW Group in Unterschleißheim, Munich.

**Conflicts of Interest:** The authors declare no conflict of interest.

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
