# Peer review of "Recycling Chain for Spent Lithium-Ion Batteries"

_metals, doi:10.3390/met10030316_

Round 1
Reviewer 1 Report
Very topical review and, as such, very welcome contribution. Also, the overview of industrial recycling technologies and operators is important. There are only some minor comments, questions and revision needs that I would like to point out.
- There are some very recent review papers on spent LIBs, as also indicated by the authors in the Introduction. However, it would be beneficial to more clearly indicate already in the first paragraph (when referring to the reviews [5&6]) the new contribution this review provides as compared to the previous reviews.
- When discussing different metallurgical options for processing spent LIBs, the pyro- and hydrometallurgical unit processes are compared as stand alone (e.g. page 8, lines 331-334: "Pyrometallurgical … high economic … complex processes … high energy consumption …"), whereas the overall process options' analyses are missing/scarce in literature. As stated by the authors, page 12 - line 411, "no company is carrying out the entire process chain". Therefore, it should be made clear that pros and cons discussed in this review are for the existing practices where optimized entire process chain options does not exist. Most probably the optimum will be found from a combination of different unit processes.
- Furthermore, the discussion seems to be entirely missing the current option of using spent LIBs (or any e-scrap) as secondary feed in existing metallurgical plants, which (totally) avoids the disadvantages of high investments. Maybe this option is not represented in the literature?
- Some detailed comments and observations:
- The use of [ ] around the reference numbers. Both {1,2,3] and [1] [2] [3] are used -> please use either or.
- Stage numbering in Figures: every fig. has "stage 1" when it should, I guess, be different; Fig. 2 "stage 1"; Fig. 3 "stage 2" etc.
- Figs. 3 & 4 have term "thermical" which should be "thermal" as in the text.
- Page 5 - line 170: "...aluminun or are..." -> something is missing?
- Page 8 - lines 313 - 314: "The metals cobalt, nickel, copper, manganese and iron an outstanding position in the recycling ..." -> Something missing?
- Page 8 - line 320: "In pyrometallurgical processes, the feed is molten and transferred into...". Normally, the feed in pyrometallurgy is solid, but in some refining unit processes it can also be molten. The sentence is not very clear and should be refined.
- Page 8 - line 322: "...graphite content of the feed has a strong influence on slag formation...". Maybe this should be explained in one more sentence, because the influence is not that self-explanatory.
- Page 8 - lines 324-326: The focus is mainly in non-ferrous metals Co, Ni, Cu, but the continued sentence gives an example of this alloy being sold as secondary raw material for alloyed steel. This is just a detail, but it would be clearer, if matte and metal alloy routes would be kept in separate sentences.
- Page 8 - line 329: "substances is technically" -> "substances from slag is technically"
- Page 12 - line 401: "Figure 5" -> "Figure 6"
- Page 12 - Fig. 6: A bit difficult to read picture. Also, too small letters in the circular parts of the boxes. The text describing the figure in lines 393-400 is difficult to perceive.
Author Response
Dear Reviewer 1,
detached in the world file, you can find my responses to your comments.
with best regards
Denis Werner

Reviewer 2 Report
The article presents a very good systematization of literature data on the recycling field of spent Li ion batteries.
Author Response
Dear Reviewer 2,
thanks for your really positive feedback. Addressing the comments from reviewer 1 and 3 marked red, I revised the article again which is detached as word file.
with best regards
Denis Werner

Reviewer 3 Report
I agree that most of the recycling technologies cannot be clearly structured, represented, compared and differentiated. Due to the increasing importance of the individual process steps prior to metallurgical treatment, this article classifies the steps for the recycling of spent lithium-ion battery along the generic process chain for waste materials and compiles an overview of the known industrial processes. This article addresses a very important issue in the field and it's well written in general. I recommend acceptance of this article after minor revisions.
1. Some terms used in the article is difficult to understand and should be replaced by more clear terms, such as "Figure 6A)-h" on line 395, "Figure 6 B-n;n>1" on line 397 and "B-p-h/B-h" on line 398, etc.
2. Typos are throughout the article and should be corrected, such as "try to categorize them according state of the art..." on line 16, "Figure 5" on line 401 should be "Figure 6", etc.
3. Some recently published research article addressing the mechanical issues in battery recycling should be cited, such as
1)Xiao, J.; Li, J.; Xu, Z. Novel Approach for in Situ Recovery of Lithium Carbonate from Spent Lithium Ion Batteries Using Vacuum Metallurgy. Environ. Sci. Technol. 2017, 51, 11960.
2) Xiao, J.; Li, J.; Xu, Z. Recycling metals from lithium ion battery by mechanical separation and vacuum metallurgy. J. Hazard. Mater. 2017, 338, 124?131.
3)Congrui Jin, Zhen Yang, Jianlin Li, Yijing Zheng, Wilhelm Pfleging, Tian Tang 2020 Bio-inspired interfaces for easy-to-recycle lithium-ion batteries. Extreme Mechanics Letters 34, 100594
Author Response
Dear Reviewer 3,
detached in the world file, you can find my responses to your comments.
with best regards
Denis Werner
